# HOUGH VOTING-BASED SELF-TRAINING FOR VISION-LANGUAGE MODEL ADAPTATION

## ABSTRACT

Traditional model adaptation framework assumes the same vocabulary across pre-training and downstream datasets, which often struggles with limited transfer flexibility and efficiency while handling downstream datasets with different vocabularies. Inspired by recent vision-language models (VLMs) that enable visual recognition defined by free-form texts via reasoning on both images and texts, we study vision-language model adaptation (VLMA), a new unsupervised model adaptation framework that positions a pre-trained VLM as the source model and transfers it towards various unlabelled downstream datasets. To this end, we propose a Hough voting-based Self-Training (HoughST) technique that introduces a multimodal Hough voting mechanism to exploit the synergy between vision and language to mitigate the distribution shift in image and text modalities simultaneously. Specifically, HoughST makes use of the complementary property of different types of features within and across vision and language modalities, which enables joint exploitation of vision and language information and effective learning of image-text correspondences in the unlabelled downstream datasets. Additionally, HoughST captures temporal information via temporal Hough voting which helps memorize and leverage previously learnt downstream dataset information. Extensive experiments show that HoughST outperforms the state-of-the-art consistently across 11 image recognition tasks. Codes will be released.

## 1 INTRODUCTION

Deep learning-based vision models He et al. (2016); Dosovitskiy et al. (2020) have achieved great success in myriad image recognition tasks but at the price of laborious annotation of large-scale training images Deng et al. (2009). To circumvent the annotation constraint, model adaptation (MA) Liang et al. (2020); Huang et al. (2021) has been explored to transfer a vision model pre-trained in certain labelled pre-training datasets towards unlabelled downstream datasets by mitigating the distribution shift in image modality. However, traditional MA Liang et al. (2020); Huang et al. (2021); Liang et al. (2021) assumes that pre-training and downstream datasets have the same vocabulary. It struggles while handling downstream datasets with different vocabularies, limiting its flexibility and efficiency greatly in unsupervised transfer.

Inspired by recent vision-language models (VLMs) Radford et al. (2021) that enable visual recognition defined by free-form texts via reasoning on both images and texts, we study vision-language model adaptation (VLMA), a new unsupervised model adaptation (UMA) framework that positions a pre-trained VLM as the source model and transfers it towards various unlabelled downstream datasets. VLMA requires a single pre-trained VLM only while transferring towards various downstream datasets of different vocabularies, instead of preparing multiple vocabulary-specific vision models with respective source datasets, as illustrated in Fig. 1. In addition, VLMA allows unsupervised transfer towards new downstream datasets with customized vocabulary, which greatly mitigates the image annotation constraint and facilitates deep network training while handling various new visual recognition tasks. On the other hand, the shift from traditional model adaptation toward VLMA comes with a new challenge, namely, the distribution shifts in both image modality and text modality.

Drawing inspiration from Hough Voting Ballard (1981); Qi et al. (2019); Lee et al. (2021) that detects a complex object by voting from its subregion information, we design Hough voting-based

Figure 1: Traditional model adaptation typically transfers a vision model across datasets of the same vocabulary, which struggles while handling downstream datasets with different vocabularies or new datasets with customized vocabularies as illustrated in (a). Inspired by the recent open-vocabulary vision-language models (VLMs), we study vision-language model adaptation, a new unsupervised model adaptation framework that positions a single pre-trained VLM as the source model and transfers it towards various unlabelled downstream datasets, offering greater transfer flexibility and efficiency, as illustrated in (b).

Self-Training (HoughST) that introduces a multimodal Hough voting mechanism to exploit the synergy between vision and language to mitigate the distribution shift in both image and text modalities simultaneously while self-training. HoughST makes use of the complementary property of different types of features within and across vision and language modalities: it exploits VLMs to encode images Lüddecke & Ecker (2022); Zang et al. (2022) and texts Lüddecke & Ecker (2022); Zang et al. (2022) into an aligned vision-language feature space and votes from the encoded visual and textual features to regularize unsupervised self-training for denoising pseudo labels and more effective self-training and vision-language model adaptation. This multimodal Hough voting mechanism enables joint exploitation of vision and language information and effective learning of image-text correspondences in the unlabelled downstream datasets. In addition, HoughST captures temporal information via temporal Hough voting, which rectifies self-training via voting from the features encoded by the intermediate models evolved along the adaptation process, ultimately helping memorize and utilize previously learnt downstream dataset information.

The proposed HoughST can be viewed as a new type of self-training with Hough voting for the task of VLMA. It has three desirable advantages: 1) it introduces visual Hough voting and textual Hough voting and enables simultaneous mitigation of distribution shift in both image and text modalities effectively; 2) it introduces temporal Hough voting along the adaptation process which allows harvesting previously learnt downstream dataset information effectively; 3) it works within an aligned image-text feature space which allows Hough voting not only within but also across vision, language and temporal dimensions, capturing their complementary advantages effectively.

In summary, the contributions of this work are threefold. *First*, we propose a novel vision-language model adaptation framework that explores Hough voting upon self-training to learn effective image-text correspondences over unlabelled downstream images. To the best of our knowledge, this is the first work that explores Hough voting for VLMA. *Second*, we design Hough voting-based self-training that introduces a multimodal Hough voting mechanism over vision, language and temporal dimensions for simultaneous mitigation of image and text distribution shift in VLMA. *Third*, extensive experiments show that the proposed Hough voting-based self-training outperforms the state-of-the-art consistently across multiple image recognition tasks.

## 2 RELATED WORK

**Model Adaptation** (MA), a type of unsupervised transfer learning, aims to adapt a model pre-trained on certain labelled pre-training datasets towards unlabelled downstream datasets. Most existing MA methods can be broadly grouped into two categories. The first category employs *generative models* to compensate for the unavailable pre-training datasets by reconstructing pre-training features Li et al. (2020); Tian et al. (2021); Qiu et al. (2021) or images Du et al. (2021); Yeh et al.

(2021); Kurmi et al. (2021); Liu et al. (2021). The second approach explores *self-training* that learns from unlabelled downstream images with predicted pseudo labels Liang et al. (2020); Huang et al. (2021); Liang et al. (2021); Xia et al. (2021); Yang et al. (2021); Ding et al. (2022b; 2023). Despite their great success, most existing methods assume the same vocabulary across the pre-training and downstream datasets and cannot handle downstream datasets with different vocabulary or new downstream dataset with customized vocabulary. This limits the flexibility and efficiency of MA greatly. We study vision-language model adaptation in this work, a new framework that reasons both images and texts and allows unsupervised transfer learning towards various unlabelled downstream datasets. We design Hough voting-based self-Training that introduces a multimodal Hough voting mechanism to explore the synergy of vision and language to mitigate image and text distribution shifts simultaneously in VLMA.

**Vision Language Model** (VLM) Radford et al. (2021); Jia et al. (2021); Yuan et al. (2021a); Yu et al. (2022); Tschannen et al. (2022); Yao et al. (2021); Wu et al. (2021); Mu et al. (2022); Cui et al. (2022); Li et al. (2021); Singh et al. (2022); Gao et al. (2022); Yang et al. (2022); Zhou et al. (2022a); Shen et al. (2022); Alayrac et al. (2022); Huang et al. (2022); Lee et al.; Chen et al. (2022b;c); Geng et al. (2023); Xu et al. (2022); Zhong et al. (2022); Li et al. (2022b); Zhao et al. (2022); Dou et al.; Yao et al. aims to learn effective vision-language correlation from image-text pairs that are almost infinitely available on the Web. It has demonstrated great potential in open-vocabulary visual recognition by recognizing images with free-form texts. On the other hand, VLMs often suffer from degraded performance due to distribution shifts with respect to various downstream datasets. Unlike recent attempts Zhou et al. (2022c;b); Yao et al. (2023); Wu et al. (2023); Khattak et al. (2022); Xing et al. (2022); Bulat & Tzimiropoulos (2022); Lu et al. (2022); Chen et al. (2022a); Ding et al. (2022a); Pratt et al. (2022); Rao et al. (2022); Yu et al. (2023) that adapt VLMs by prompt tuning with few-shot downstream dataset images, we focus on adapting VLMs towards various downstream datasets by ingeniously exploiting the unlabelled images which are often off-the-shelf available in abundance.

**Hough Voting** detects complex objects by aggregating votes from their subregions and surrounding areas, leveraging spatially complementary information to enhance vision tasks. Existing methods can be broadly classified into two categories. The first category is classical Hough voting, which relies on traditional visual patterns. For example, Ballard (1981) detects the presence of complex objects by voting from image patches, Leibe et al. (2008) proposes the implicit shape model, Sun et al. (2010) integrates depth information into Hough voting, Maji & Malik (2009) designs importance-aware voting, and Gall et al. (2011); Gall & Lempitsky (2013) develop Hough forests. The second category is deep Hough voting, which incorporates voting mechanisms into deep neural networks. For instance, Kehl et al. (2016) uses deep features for 6D pose estimation, Milletari et al. (2017) learns deep features to build codebooks, and Qi et al. (2019); Lee et al. (2021) apply Hough voting within deep networks for 3D learning. In contrast to previous approaches, we propose HoughST that works within an aligned image-text feature space which enables Hough voting not only within but also across visual, language and temporal dimensions, effectively capturing their complementary strengths for vision-language model adaptation.

## 3 METHOD

### 3.1 PRELIMINARIES OF VISION-LANGUAGE MODEL

**Vision-language model (VLM) training.** VLM Radford et al. (2021); Jia et al. (2021); Yuan et al. (2021a); Yu et al. (2022); Tschannen et al. (2022) learns effective vision-language correlation from image-text pairs that are almost infinitely available on the Web Radford et al. (2021); Schuhmann et al. (2021). The training involves a VLM $F = \{F^I, F^T\}$ where $F^I$ and $F^T$ denote an image encoder and a text encoder respectively, and an image-text dataset $D_s = \{(x_n^I, x_n^T)\}_{n=1}^N$ where $x_n^I$ and $x_n^T$ stand for an image sample and its paired text sample. Given $F$ and $D_s$, rich vision-language correlation can be learnt with a vision-language training objective such as image-text contrast Radford et al. (2021) as follows:

$$\mathcal{L}_{\text{VLM}} = -\sum_{i=1}^N \log \frac{\exp\left(z_i^I \cdot z_i^T / \tau\right)}{\sum_{j=1}^N \exp(z_i^I \cdot z_j^T / \tau)} - \sum_{i=1}^N \log \frac{\exp\left(z_i^T \cdot z_i^I / \tau\right)}{\sum_{j=1}^N \exp(z_i^T \cdot z_j^I / \tau)}, \quad (1)$$

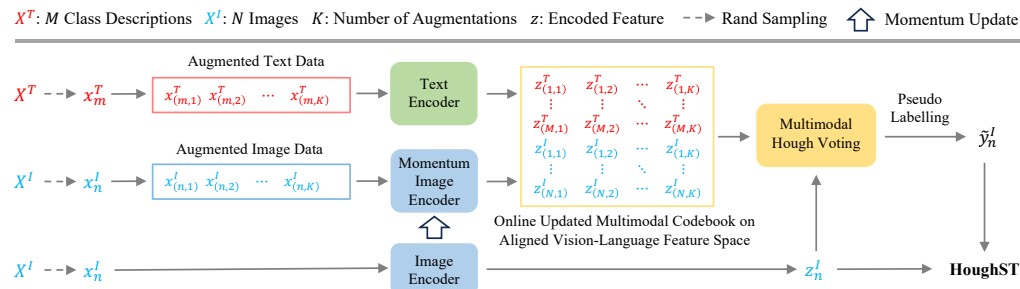

Figure 2: **Overview of HoughST.** HoughST encodes texts and images into an aligned vision-language feature space and votes from the encoded visual and textual features (i.e., Multimodal Codebook) to regularize unsupervised self-training, enabling joint exploitation of vision and language information and effective learning of image-text correspondences in the unlabelled downstream datasets. In addition, HoughST updates Multimodal Codebook online using the features encoded by the intermediate models evolved along the adaptation process, enabling temporal Hough voting and helping memorize and utilize previously learnt downstream dataset information.

where the two terms on the right denote image-to-text and text-to-image contrastive losses respectively. The notations $z_i^I = F^I(x_i^I)$ and $z_i^T = F^T(x_i^T)$ stand for the encoded image and text features respectively, $\tau$ denotes a temperature parameter Wu et al. (2018), and "·" stands for the inner-product that measures the cosine similarity between two features.

**VLM inference.** A pre-trained VLM can perform open-vocabulary image recognition on various unlabelled downstream datasets by reasoning on both images and texts Radford et al. (2021). Given an unlabelled downstream dataset $D = \{X^I, X^T\}$, $X^I = \{x_n^I\}_{n=1}^N$ stands for $N$ unlabelled images and $X^T = \{x_m^T\}_{m=1}^M$ denotes $M$ class names of interest, e.g., $X^T = \{$car, bus, ..., bike, person$\}$. The pre-trained VLM predicts the probability of an image $x^I$ belonging to class $x^T$ by:

$$p_{x^I \to x^T} = z^I \cdot z^T, \tag{2}$$

where $z^I = F^I(x^I)$, $z^T = F^T(x^T)$. Theoretically, VLMs can work with class names $X^T$ defined by free-form texts and thus achieve open-vocabulary image recognition. Note $X^T = \{x_m^T\}_{m=1}^M$ contains $M$ downstream-dataset class names but provides no information of which image belongs to which class name Radford et al. (2021).

**Distribution shifts lead to degraded performance.** VLMs often suffer from degraded performance due to distribution shifts with respect to various downstream datasets Li et al. (2022a). For example, for distribution shifts in text modalities, VLMs are largely pre-trained on the pre-training datasets that consist of free-form sentences while the downstream datasets generally provide only raw class names, where such distribution shifts between pre-training and downstream datasets often lead to degraded performance. For distribution shifts in image modalities, VLMs are largely pre-trained on normal images from the internet while downstream datasets may have quite different distributions, e.g., images in synthetic, Clipart, Sketch styles etc., where such distribution shifts usually lead to degraded performance. Previous works Radford et al. (2021); Zhou et al. (2022c); Li et al. (2022a); Bahng et al. (2022) also show that there are little overlap between the VLM training data and the testing downstream data, and properly tackle the gaps between them via text or visual prompt learning or model finetuning could improve the performance on downstream datasets.

### 3.2 DEFINITION OF VISION-LANGUAGE MODEL ADAPTATION (VLMA)

This work focuses on the task of VLMA, a new unsupervised model adaptation (UMA) framework that transfers a pre-trained VLM $F = \{F^I, F^T\}$ towards an unlabelled downstream dataset $D = \{X^I, X^T\}$ with certain unsupervised training losses, i.e., $\mathcal{L}_{\text{VLMA}} = \mathcal{L}_{\text{unsupervised}}(X^I, X^T; F^I, F^T)$. Take self-training Zhu (2005); Zou et al. (2018) as an example. Given $X^I = \{x_n^I\}_{n=1}^N$ and $X^T = \{x_m^T\}_{m=1}^M$, the unsupervised training loss on unlabelled downstream data can be formulated as the following:

$$\hat{y}_n^I = \arg\max_m \ z_n^I \cdot z_m^T, \ \ \mathcal{L}_{\text{ST}} = -\sum_{n=1}^{N} \log \frac{\sum_{m=1}^{M} \exp\left(z_n^I \cdot z_m^T / \tau\right) \times \mathbb{1}(\hat{y}_n^I == m)}{\sum_{m=1}^{M} \exp(z_n^I \cdot z_m^T / \tau)}, \tag{3}$$

where $z_n^I$ and $z_m^T$ denote the encoded image and text features, i.e., $z_n^I = F^I(x_n^I)$ and $z_m^T = F^T(x_m^T)$. $\hat{y}_n^I$ stands for the pseudo label of $x_n^I$.

Note the unsupervised training is often unstable and susceptible to collapse if we optimize VLM image encoder and text encoder concurrently Li et al. (2022a). Hence, we freeze the VLM text encoder during unsupervised model adaptation for stable adaptation.

### 3.3 HOUGH VOTING-BASED SELF-TRAINING

We tackle the challenge of VLMA from a perspective of Hough Voting Ballard (1981); Qi et al. (2019); Lee et al. (2021). As illustrated in Fig. 2, we design Hough voting-based Self-Training (HoughST) that introduces visual Hough voting and textual Hough voting over self-training to mitigate the distribution shifts in image and text modalities simultaneously. In addition, HoughST captures temporal information via temporal Hough voting, which rectifies self-training via voting from the features encoded by the intermediate models evolved along the adaptation process, ultimately helping memorize and utilize previously learnt downstream dataset information.

**Textual Hough voting** gathers the text features encoded from different types of text descriptions for Hough voting, aiming to leverage the complementary information of various text descriptions (i.e., different types of text descriptions for a class Lüddecke & Ecker (2022); Zang et al. (2022)) to mitigate the distribution shift in text modality. It employs a Large Language Model (LLM) Brown et al. (2020); Wang & Komatsuzaki (2021) to generate different types of text descriptions for a given class name and then encodes them by the VLM text encoder. The encoded text features are then fused in a two-step manner: 1) uniformly average the multiple text features to acquire an initial voting centroid; 2) calculate the final voting centroid by weighted average where the weight of each feature is the distance between it and the initial voting centroid. This two-step voting operation allows smooth feature fusion by weighting down the effect of corner cases, which is important for textual Hough voting as the LLM-generated text descriptions are not always reliable (e.g., when experiencing generation failures, LLM may generate only a full stop character "." or a random word).

Given a class name $x_m^T \in X^T$, we employ the Large Language Model Brown et al. (2020) to generate $K$ text descriptions $\{x_{(m,1)}^T, x_{(m,2)}^T, ..., x_{(m,K)}^T\}$ and then the VLM text encoder $F^T$ to encode the generated text descriptions to acquire text features $\{z_{(m,1)}^T, z_{(m,2)}^T, \ ... \ , z_{(m,K)}^T\}$ (i.e., $z_{(m,k)}^T = F^T(x_{(m,k)}^T)$). The text features are then fused in a two-step voting manner to get the final textual Hough voting centroid $\delta_m^T$:

$$\delta_m^{T_{\text{initial}}} = \frac{1}{K} \sum_{k=1}^{K} z_{(m,k)}^T, \ \ \delta_m^T = \sum_{k=1}^{K} (z_{(m,k)}^T \cdot \delta_m^{T_{\text{initial}}}) \times z_{(m,k)}^T, \tag{4}$$

where "·" denotes inner-product and $(z_{(m,k)}^T \cdot \delta_m^{T_{\text{initial}}})$ measures the distance between $z_{(m,k)}^T$ and $\delta_m^{T_{\text{initial}}}$.

**Visual Hough voting** gathers the image features encoded from different images of the same category for Hough voting, aiming to utilize the complementary information of different types of images (i.e., various images as the visual descriptions for a class Lüddecke & Ecker (2022); Zang et al. (2022)) for mitigating the distribution shift in image modality. Given an image, it employs certain off-the-shelf image augmentation policies Cubuk et al. (2020) to generate multiple image augmentations, encodes them with the VLM image encoder, and fuses the encoded image features in a class-wise manner. Since downstream images are unlabelled, we generated pseudo labels for class-wise image feature fusion. The class-wise feature fusion allows category-wise image information consolidation, which is crucial to visual Hough voting due to the abundance of downstream dataset images and the encoded image features. In addition, it simplifies vision-language Hough voting greatly (described in the later paragraphs) as textual Hough voting naturally works in a category-wise manner. Besides, with temporal Hough voting (described in the later paragraphs), it allows to dynamically select image features using pseudo labels along the adaptation process to describe each class visually.

Given an image $x_n^I \in X^I$, we adopt the off-the-shelf image augmentation policies in Cubuk et al. (2020) to generate $K$ image augmentations $\{x_{(n,1)}^I, x_{(n,2)}^I, ..., x_{(n,K)}^I\}$ and then the VLM image encoder $F^I$ to encode the generated image data to acquire image features $\{z_{(n,1)}^I, z_{(n,2)}^I, ..., z_{(n,K)}^I\}$ (i.e., $z_{(n,k)}^I = F^I(x_{(n,k)}^I)$). Finally, the encoded features are fused in a class-wise voting manner to get the visual Hough voting centroid $\delta_m^I$:

$$\delta_m^I = \frac{\sum_n^N \sum_{k=1}^K z_{(n,k)}^I \times \mathbb{1}(\hat{y}_{(n,k)}^I == m)}{\sum_n^N \sum_{k=1}^K \mathbb{1}(\hat{y}_{(n,k)}^I == m)}, \tag{5}$$

where $\mathbb{1}(\hat{y}_{(n,k)}^I == m)$ returns "1" if $\hat{y}_{(n,k)}^I = m$ else 0. Note $\hat{y}_{(n,k)}^I = \arg\max_m z_{(n,k)}^I \cdot z_m^T$ denotes the pseudo label of $x_{(n,k)}^I$. Note we employ the momentum update of $F^I$ in the vision feature voting for stable feature encoding and better capturing of temporal information as in Fig. 2.

**Temporal vision-language Hough voting** exploits the synergy between vision and language by gathering different types of text features and image features over an aligned vision-language feature space. It employs the textual and visual Hough voting centroids as starting point and updates them with the visual Hough voting centroids generated by the intermediate VLM image encoder evolved along the adaptation process. This enables Hough voting not only within but also across vision and language modalities, capturing the complementary advantages of vision and language information effectively. In addition, the updating also achieves **temporal Hough voting** that gathers and leverages previously learnt downstream dataset information effectively. Note we conduct temporal Hough voting from image features only as the VLM text encoder is frozen during the adaptation process.

Specifically, we use the textual and visual Hough voting centroids $\delta_m^T$ and $\delta_m^I$ to initialize the vision-language Hough voting centroid $\delta_m^{IT}$ and keep updating $\delta_m^{IT}$ with $\delta_m^I$ along the adaptation process as follows:

$$\delta_m^{IT_{\text{initial}}} = \delta_m^I + \delta_m^T, \quad \delta_m^{IT*} \leftarrow \lambda \delta_m^{IT} + (1 - \lambda)\delta_m^I, \tag{6}$$

where $\delta_m^{IT}$ and $\delta_m^{IT*}$ denote the vision-language Hough voting centroid before and after one update, respectively. $\lambda$ is a coefficient that controls the update speed of temporal Hough voting. Note the first part denotes *vision-language Hough voting* while the second part denotes *temporal Hough voting*.

**Hough voting-based self-training.** Given vision-language Hough voting centroid $\delta_m^{IT}$, downstream images, $X^I = \{x_n^I\}_{n=1}^N$ and downstream class names $X^T = \{x_m^T\}_{m=1}^M$, we employ $\delta_m^{IT}$ to vote to regularize unsupervised self-training, which can be formulated as follows:

$$\tilde{y}_n^I = \arg\max_m \ (z_n^I \cdot z_m^T) \times (z_n^I \cdot \delta_m^{IT}), \tag{7}$$

$$\mathcal{L}_{\text{HoughST}} = -\sum_{n=1}^N \log \frac{\sum_{m=1}^M \exp(z_n^I \cdot z_m^T / \tau) \times \mathbb{1}(\tilde{y}_n^I == m)}{\sum_{m=1}^M \exp(z_n^I \cdot z_m^T / \tau)}, \tag{8}$$

where $z_n^I$ and $z_m^T$ denote the encoded image and text features, i.e., $z_n^I = F^I(x_n^I)$ and $z_m^T = F^T(x_m^T)$. $\tilde{y}_n^I$ stands for the pseudo label of $x_n^I$ generated with $\delta_m^{IT}$. The vision-language Hough voting centroid $\delta_m^{IT}$ captures rich downstream image and text information. It is thus more invariant to visual and textual distribution shifts and can vote from the captured information to regularize self-training to generate more accurate pseudo labels.

# 4 EXPERIMENTS

## 4.1 IMPLEMENTATION DETAILS

We conduct experiments with three popular backbones, i.e., ResNet-50 He et al. (2016), ResNet-101 He et al. (2016) and ViT-B Dosovitskiy et al. (2020) pre-trained by CLIP Radford et al. (2021). In training, we employ AdamW optimizer Loshchilov & Hutter (2017) with a weight decay of $0.05$, and set the initial learning rate as $1e - 5$ which is adjusted with a cosine learning rate schedule. We use 2 GPUs with batch size $64$ and the unsupervised adaptation training adds only a small amount of computation overhead after VLM pre-training. We set input image size as $224 \times 224$ and employ data augmentation policies of "RandomResizedCrop+Flip+RandAug" Cubuk et al. (2020) for image data

Table 1: VLMA performance on multi-style datasets of Office, Office-Home and Adaptiope. For fair comparisons, the results of all methods are based on the baseline CLIP.

| ViT-B/16 | Office | | | | | Office-Home | | | | | Adaptiope | | | |
|---|---|---|---|---|---|---|---|---|---|---|---|---|---|---|
| | A | W | D | S | Mean | A | C | P | R | Mean | P | R | S | Mean |
| CLIP (baseline) | 77.9 | 79.4 | 76.9 | 56.7 | 72.7 | 74.4 | 58.5 | 79.6 | 79.4 | 72.9 | 82.6 | 78.2 | 45.9 | 68.9 |
| ST | 78.6 | 81.1 | 78.3 | 68.6 | 76.6 | 77.8 | 62.5 | 81.3 | 80.3 | 75.4 | 86.7 | 82.0 | 49.5 | 72.7 |
| CBST Zou et al. (2018) | 79.1 | 80.7 | 78.5 | 68.9 | 76.8 | 77.3 | 62.8 | 81.7 | 80.7 | 75.6 | 86.9 | 83.2 | 50.1 | 73.4 |
| CRST Zou et al. (2019) | 78.8 | 81.2 | 79.1 | 69.0 | 77.0 | 78.1 | 63.1 | 81.4 | 81.1 | 75.9 | 87.1 | 83.9 | 50.7 | 73.9 |
| SHOT Liang et al. (2020) | 79.2 | 81.1 | 81.2 | 67.1 | 77.1 | 77.9 | 64.3 | 80.9 | 81.5 | 76.1 | 88.3 | 84.7 | 51.2 | 74.7 |
| MUST Li et al. (2022a) | 79.0 | 81.4 | 79.5 | 69.2 | 77.2 | 77.7 | 63.9 | 82.1 | 81.4 | 76.2 | 88.8 | 85.3 | 51.5 | 75.2 |
| HoughST (Ours) | **84.3** | **82.8** | **81.3** | **72.3** | **80.1** | **78.9** | **68.9** | **85.7** | **82.4** | **78.9** | **91.8** | **88.1** | **59.8** | **79.9** |

| ResNet-50 | Office | | | | | Office-Home | | | | | Adaptiope | | | |
|---|---|---|---|---|---|---|---|---|---|---|---|---|---|---|
| | A | W | D | S | Mean | A | C | P | R | Mean | P | R | S | Mean |
| CLIP (baseline) | 72.9 | 68.9 | 73.1 | 48.2 | 65.7 | 64.6 | 42.1 | 71.9 | 71.9 | 62.6 | 74.5 | 66.2 | 35.8 | 58.8 |
| ST | 75.2 | 66.8 | 71.3 | 44.1 | 64.3 | 66.7 | 38.6 | 72.0 | 73.8 | 62.7 | 75.7 | 70.7 | 26.7 | 57.7 |
| CBST Zou et al. (2018) | 75.2 | 67.8 | 72.2 | 51.1 | 66.5 | 68.1 | 41.5 | 73.6 | 74.5 | 64.4 | 77.2 | 71.1 | 34.3 | 60.8 |
| CRST Zou et al. (2019) | 76.4 | 67.4 | 74.5 | 52.3 | 67.6 | 68.3 | 42.3 | 74.8 | 75.3 | 65.1 | 78.3 | 71.2 | 36.2 | 61.9 |
| SHOT Liang et al. (2020) | 77.5 | 70.1 | 76.8 | 54.8 | 69.8 | 68.4 | 44.2 | 75.7 | 75.6 | 65.9 | 78.5 | 72.4 | 36.8 | 62.5 |
| HoughST (Ours) | **79.6** | **75.3** | **80.3** | **55.0** | **72.5** | **68.6** | **47.9** | **78.2** | **77.4** | **68.0** | **80.7** | **75.6** | **37.8** | **64.7** |

| ResNet-101 | Office | | | | | Office-Home | | | | | Adaptiope | | | |
|---|---|---|---|---|---|---|---|---|---|---|---|---|---|---|
| | A | W | D | S | Mean | A | C | P | R | Mean | P | R | S | Mean |
| CLIP (baseline) | 73.2 | 73.8 | 75.1 | 50.2 | 68.0 | 69.5 | 47.8 | 74.3 | 74.2 | 66.4 | 75.9 | 69.0 | 35.3 | 60.0 |
| ST | 74.4 | 74.2 | 73.8 | 54.3 | 69.1 | 71.4 | 43.2 | 74.9 | 75.0 | 66.1 | 78.4 | 71.8 | 37.8 | 62.6 |
| CBST Zou et al. (2018) | 74.6 | 75.9 | 72.9 | 58.1 | 70.3 | 72.3 | 44.9 | 77.7 | 76.2 | 67.7 | 79.5 | 73.3 | 41.5 | 64.7 |
| CRST Zou et al. (2019) | 75.3 | 76.6 | 73.4 | 58.5 | 70.9 | 73.4 | 45.9 | 78.4 | 76.8 | 68.6 | 80.1 | 75.2 | 43.7 | 66.3 |
| SHOT Liang et al. (2020) | 76.9 | 78.2 | 75.1 | 59.0 | 72.3 | 73.5 | 47.2 | 79.1 | 77.4 | 69.3 | 81.9 | 76.3 | 44.1 | 67.4 |
| HoughST (Ours) | **80.1** | **81.2** | **77.5** | **61.9** | **75.1** | **74.6** | **51.2** | **82.6** | **78.9** | **71.8** | **85.3** | **78.8** | **45.7** | **69.9** |

augmentation. The momentum VLM image encoder $F_m^I$ is updated with a momentum coefficient of 0.99, i.e., $\theta_{F_m^I} \leftarrow \gamma\, \theta_{F_m^I} + (1-\gamma)\theta_{F^I}$, and $\gamma$ is a momentum coefficient. All results except on ImageNet are obtained with above implementation details. For the large-scale ImageNet, we follow the implementations in Li et al. (2022a) and use 16 GPUs with batch size 1024. During evaluation, we simply use the center-cropped image.

## 4.2 HoughST on Multi-style Datasets

Tables 1-3 report the image classification results on 4 representative multi-style datasets. The experiments were conducted with 3 representative backbones, i.e., ResNet-50, ResNet-101 and ViT-B/16. It can be seen that our HoughST achieves superior performance consistently over various styles as compared with state-of-the-art methods. Besides, HoughST outperforms CLIP substantially on Office (S)ynthetic style, Office-Home (C)lipart style and Adaptiope (S)ynthetic style with 15.6%, 10.4% and 13.9% accuracy improvement, respectively, showing that HoughST can well handle the downstream datasets with large distribution shifts, i.e., Synthetic and Clipart styles.

## 4.3 HoughST on Task-specific Datasets

Table 4 reports the image classification over 5 popular task-specific datasets as in prior work Li et al. (2022a). The experiments were conducted with 3 representative backbones, i.e., ResNet-50, ResNet-101 and ViT-B/16 (the results with ResNet-101 are provided in the appendix). We can observe that HoughST outperforms the state-of-the-arts by large margins consistently over different task-specific datasets, demonstrating that it can effectively handle various new visual recognition tasks by using unlabelled data. In addition, HoughST brings substantial improvements upon CLIP over SUN397 (e.g., +11.0% on ViT-B/16) and GTSRB (e.g., +16.8% on ViT-B/16), showing that HoughST can well tackle new image classification tasks with very specific objectives, e.g., indoor/outdoor scene and German traffic sign recognition.

## 4.4 VLMA on General Dataset ImageNet

Table 5 presents ImageNet results. It can be seen that HoughST achieves superior performance as compared with state-of-the-art unsupervised methods, demonstrating its effectiveness over the very diverse and large-scale ImageNet. Besides, introducing our HoughST into 16-shot supervised meth-

Table 2: VLMA performance on large-scale multi-style dataset VisDA. For fair comparisons, the results of all methods are based on the baseline CLIP.

| VisDA Synthesis Style | | | | | | | | | | | | |
|---|---|---|---|---|---|---|---|---|---|---|---|---|
| ViT-B/16 | plane | bcycl | bus | car | horse | knife | mcycl | person | plant | sktbrd | train | truck | Per-class |
| CLIP (baseline) | 98.5 | 99.7 | 64.6 | 92.5 | 99.7 | 96.8 | 85.3 | 98.4 | 99.8 | 79.4 | 66.4 | 73.4 | 87.8 |
| ST | 97.2 | **99.9** | 60.4 | 84.5 | 99.8 | 98.6 | 92.5 | 99.7 | **99.9** | 79.3 | 74.2 | 84.4 | 89.2 |
| CBST Zou et al. (2018) | 98.4 | 99.7 | 67.3 | 85.2 | 99.8 | 99.1 | 95.3 | **99.9** | 99.4 | 83.4 | 83.4 | 87.4 | 91.5 |
| CRST Zou et al. (2019) | 98.1 | 98.2 | 70.5 | 86.5 | 98.6 | 98.7 | 94.3 | 98.8 | 97.8 | 86.7 | 88.7 | 86.1 | 91.9 |
| SHOT Liang et al. (2020) | 99.6 | 99.1 | 74.6 | 86.3 | 98.3 | **99.3** | **96.4** | 96.1 | 99.7 | 87.5 | 90.1 | 87.3 | 92.2 |
| MUST Li et al. (2022a) | 98.7 | 99.2 | 76.3 | 86.4 | 99.6 | 99.2 | 95.3 | 99.3 | 99.8 | 89.2 | 89.9 | 82.6 | 92.9 |
| HoughST (Ours) | **99.7** | 99.7 | **78.9** | **86.6** | **99.9** | **99.3** | **96.4** | 99.4 | 99.8 | **91.9** | **90.8** | **93.2** | **94.6** |

| VisDA Real Style | | | | | | | | | | | | |
|---|---|---|---|---|---|---|---|---|---|---|---|---|
| ViT-B/16 | plane | bcycl | bus | car | horse | knife | mcycl | person | plant | sktbrd | train | truck | Per-class |
| CLIP (baseline) | 98.9 | 91.0 | 90.5 | 65.7 | 98.6 | 89.1 | 95.3 | 56.5 | 90.2 | 96.8 | 93.8 | **75.8** | 86.8 |
| ST | **99.4** | 87.3 | 92.5 | 68.3 | 98.7 | 90.4 | 94.6 | 69.3 | 91.2 | 96.7 | 94.5 | 66.4 | 87.3 |
| CBST Zou et al. (2018) | 99.3 | 89.2 | 91.3 | **76.9** | 98.2 | 89.5 | 95.4 | 68.1 | 88.4 | 96.4 | 94.1 | 64.2 | 87.5 |
| CRST Zou et al. (2019) | 99.1 | 90.7 | 91.4 | 64.5 | 99.1 | 93.4 | 95.1 | 68.2 | 91.3 | 96.8 | 95.3 | 67.2 | 87.6 |
| SHOT Liang et al. (2020) | 99.3 | 92.8 | 91.9 | 65.3 | 98.7 | 95.2 | 94.5 | 67.7 | 92.1 | 96.9 | 95.4 | 67.9 | 88.1 |
| MUST Li et al. (2022a) | 99.2 | 95.7 | **92.6** | 56.9 | 99.1 | **98.6** | 96.0 | 67.0 | **93.5** | **98.8** | **96.9** | 68.1 | 88.5 |
| HoughST (Ours) | 99.2 | **95.9** | 92.1 | 66.1 | **99.2** | 97.8 | **96.7** | **70.8** | 92.7 | 98.4 | 96.2 | 74.6 | **90.0** |

Table 3: VLMA performance on multi-style datasets of DomainNet. For fair comparisons, the results of all methods are based on the baseline CLIP.

| Method | ViT-B/16 | | | | | | | ResNet-50 | | | | | | |
|---|---|---|---|---|---|---|---|---|---|---|---|---|---|---|
| | Clipart | Info | Paint | Quick | Real | Sketch | Mean | Clipart | Info | Paint | Quick | Real | Sketch | Mean |
| CLIP (baseline) | 69.7 | 47.8 | 65.0 | 14.5 | 82.0 | 62.4 | 56.9 | 51.9 | 39.1 | 52.1 | 6.4 | 74.7 | 47.4 | 45.3 |
| ST | 72.5 | 51.3 | 68.7 | 12.4 | 83.7 | 64.3 | 58.8 | 55.4 | 40.5 | 54.8 | 4.3 | 76.2 | 48.3 | 46.5 |
| CBST Zou et al. (2018) | 74.3 | 56.8 | 69.8 | 13.4 | 83.1 | 67.1 | 60.7 | 56.3 | 40.7 | 56.2 | 5.6 | 77.4 | 48.1 | 47.3 |
| CRST Zou et al. (2019) | 75.6 | 56.9 | 71.3 | 14.8 | 83.3 | 68.2 | 61.7 | 57.9 | 41.8 | 57.1 | 6.2 | 78.2 | 49.5 | 48.4 |
| SHOT Liang et al. (2020) | 75.9 | 57.4 | 71.5 | 15.1 | 83.3 | 68.8 | 62.0 | 60.3 | 45.8 | 60.5 | 5.1 | 78.9 | 54.1 | 50.8 |
| MUST Li et al. (2022a) | 76.1 | 57.5 | 71.6 | 14.2 | 84.4 | 68.9 | 62.1 | - | - | - | - | - | - | - |
| HoughST (Ours) | **77.6** | **59.0** | **73.1** | **18.2** | **86.1** | **70.1** | **64.0** | **62.7** | **47.2** | **61.3** | **7.2** | **80.2** | **54.4** | **52.2** |

ods further improves the performance clearly, showing that 16-shot supervised and our unsupervised methods are complementary to each other as they focus on exploring different types of data.

### 4.5 DISCUSSION

**Ablation study.** We conduct ablation studies with ViT-B/16 on Office as shown in Table 6. As the core of the proposed HoughST, we examine how our designed *visual Hough voting*, *textual Hough voting* and *temporal Hough voting* contribute to the overall performance of vision-language model adaptation. As shown in Table 6, including either visual Hough voting or textual Hough voting above self-training improves performance clearly, showing that voting from different types of image/text features help mitigate distribution shifts in image modality/text modality and can regularize unsupervised self-training with more accurate pseudo label prediction. In addition, combining visual and textual Hough voting performs clearly better, indicating that the two types of Hough voting complement each other by working from orthogonal vision and language perspectives. Furthermore, including *temporal Hough voting* upon vision-language Hough voting, i.e., HoughST in the last row, performs the best. It demonstrates the importance of temporal Hough voting that helps memorize and leverage previously learnt downstream datasets information along the training process.

**Parameter study.** The parameter $\lambda$ in Eq. 6 controls the update speed of temporal information fusion and voting. We investigate $\lambda$ by varying it from 0.9 to 0.9999 progressively, as shown in Table 8. It can be seen that varying $\lambda$ does not affect HoughST clearly. The performance drops a bit while $\lambda = 0.9$, largely because a fast update may lead to unstable temporal information fusion and voting which only captures local information within each training batch.

**Comparison with other voting methods.** We compare HoughST with other voting strategies that explore complementary advantages of different features via uniform voting Jiang et al. (2020); Schick & Schütze (2020); Yuan et al. (2021b), weighted voting Jiang et al. (2020); Qin & Eisner (2021); Schick & Schütze (2020), majority voting Lester et al. (2021); Hambardzumyan et al. (2021). As Table 7 shows, existing voting methods do not perform well, largely because they were designed for a single data modality without considering (1) the joint exploitation of vision and lan-

Table 4: VLMA performance on task-specific datasets of various image recognition tasks. For fair comparisons, the results of all methods are based on the baseline CLIP.

| Method | ViT-B | | | | | | ResNet-50 | | | | | |
|---|---|---|---|---|---|---|---|---|---|---|---|---|
| | SUN397 | Food101 | GTSRB | DTD | UCF101 | Mean | SUN397 | Food101 | GTSRB | DTD | UCF101 | Mean |
| CLIP (baseline) | 60.8 | 85.6 | 32.5 | 44.5 | 64.1 | 57.5 | 54.0 | 73.1 | 25.0 | 39.8 | 56.0 | 49.5 |
| ST | 65.8 | 88.2 | 32.8 | 45.0 | 67.0 | 59.7 | 59.0 | 74.4 | 20.5 | 35.8 | 56.4 | 49.2 |
| CBST Zou et al. (2018) | 63.2 | 89.5 | 37.6 | 44.3 | 68.1 | 60.5 | 63.7 | 78.2 | 27.4 | 38.7 | 59.5 | 53.5 |
| CRST Zou et al. (2019) | 64.7 | 89.1 | 39.7 | 45.3 | 68.6 | 61.4 | 64.2 | 76.5 | 30.1 | 39.4 | 61.3 | 54.3 |
| SHOT Liang et al. (2020) | 66.1 | 89.6 | 41.2 | 46.3 | 69.4 | 62.5 | 65.1 | 77.3 | 34.6 | 41.2 | 62.7 | 56.1 |
| MUST Li et al. (2022a) | 67.7 | 89.4 | 42.7 | 46.5 | 70.6 | 63.3 | - | - | - | - | - | - |
| HoughST (Ours) | **71.8** | **91.1** | **49.3** | **52.7** | **73.9** | **67.7** | **65.7** | **79.5** | **39.6** | **49.4** | **65.6** | **59.9** |

Table 5: Comparison with few-shot supervised adaptation methods and unsupervised adaption methods on ImageNet. All methods use the same CLIP ViT-B/16 model as the baseline.

| Method | CLIP | Supervised with 16 Labels per Class | | | | Unsupervised | | |
|---|---|---|---|---|---|---|---|---|
| | | CoCoOp | CoOp | CoOp + HoughST | CoCoOp + HoughST | ST | MUST | HoughST (Ours) |
| ImageNet | 68.3 | 71.0 | 71.5 | 79.6 | 79.8 | 76.5 | 77.7 | **78.7** |

Table 6: Ablation studies of HoughST with ViT-B/16 on Office dataset.

| Method | Vision-Language Hough voting | | Temporal Hough voting | Office (Mean) |
|---|---|---|---|---|
| | Visual Hough voting | Textual Hough voting | | |
| CLIP (baseline) | | | | 72.7 |
| ST | | | | 76.6 |
| | ✓ | | | 77.5 |
| | | ✓ | | 78.2 |
| | ✓ | ✓ | | 78.7 |
| HoughST | ✓ | ✓ | ✓ | **80.1** |

Table 7: Comparison with other voting methods with ViT-B/16 on Office.

| Method | Office (Mean) |
|---|---|
| ST + Importance-aware Voting Maji & Malik (2009) | 77.3 |
| ST + Uniform Voting Jiang et al. (2020) | 77.2 |
| ST + Weighted Voting Qin & Eisner (2021) | 77.4 |
| ST + Majority Voting Lester et al. (2021) | 77.0 |
| HoughST (Ours) | **80.1** |

Table 8: Parameter ablations with ViT-B/16 on Office. The default is marked in gray .

| Parameter $\lambda$ | 0.9 | 0.99 | 0.999 | 0.9999 |
|---|---|---|---|---|
| Office (Mean) | 79.6 | **80.1** | 80.1 | 80.0 |

guage modalities and (2) the information memorization during unsupervised transfer. HoughST instead learns and memorizes effective image-text correspondences in the unlabelled downstream datasets via joint exploitation of vision and language information, which are essential to vision-language model adaptation.

**Pseudo label accuracy.** Fig. 3 shows the pseudo label accuracy along the unsupervised adaptation process. HoughST generates much more accurate pseudo labels than the vanilla self-training (ST) and the state-of-the-art MUST. The superior pseudo label accuracy is largely attributed to the proposed multimodal Hough voting which helps capture rich downstream dataset image and text information that is more invariant to visual and textual distribution shifts and can better regularize unsupervised self-training.

**Visualization of multimodal Hough voting.** We analyze how our proposed multimodal Hough voting mechanisms work by visualizing the feature distribution, as shown in Figure 4. From Figure 4

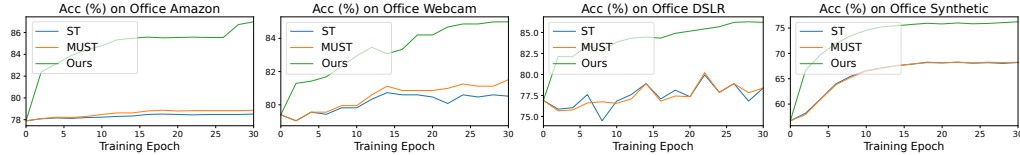

Figure 3: Pseudo label accuracy along the unsupervised adaptation process (with ViT-B/16).

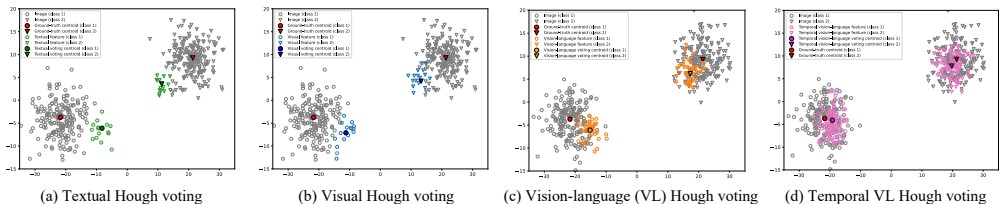

(a) Textual Hough voting  (b) Visual Hough voting  (c) Vision-language (VL) Hough voting  (d) Temporal VL Hough voting

Figure 4: Visualization of multimodal Hough voting. It shows that all three voting mechanisms in our HoughST can capture different types of image and text features, which build an informative, up-to-date and accurate multimodal codebook for Hough voting, ultimately voting together to produce better voting centroids (i.e., closer to ground-truth centroids) and VLMA performance.

(a), (b) and (c), we can observe that textual Hough voting and visual Hough voting can capture different types of image and text features respectively, which complement each other and provide orthogonal vision and language information for more comprehensive voting. In addition, Figure 4 (d) shows that including temporal Hough voting further enriches the distribution of vision-language feature, which helps build an informative, up-to-date and accurate multimodal codebook for Hough voting, leading to better voting centroids that are closer to ground-truth centroids and facilitating vision-language model adaptation.

Due to the space limit, we provide more dataset details, experiments and discussions in the appendix.

## 5 CONCLUSION

This paper presents HoughST, a novel vision-language model adaptation framework that explores Hough voting to learn effective image-text correspondences over unlabelled downstream dataset images. HoughST introduces a multimodal Hough voting mechanism over vision, language and temporal dimensions for simultaneous mitigation of image and text distribution shifts in VLMA. It requires merely a single pre-trained VLM but achieves effective and efficient unsupervised model adaptation towards various unlabelled downstream datasets, demonstrating its superiority in facilitating deep network training while handling various new visual recognition tasks and styles. Extensive experiments show that HoughST achieves superb recognition performance consistently across different backbones and image recognition tasks and styles. Moving forward, we will explore HoughST for other vision tasks such as segmentation and detection.

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

## A   APPENDIX

We provide dataset details in Section B, full comparisons with the state-of-the-art methods in Section C and pseudo codes of the proposed HoughST in Section D. In addition, we provide more discussion experiments, including the analysis of our proposed Textual Hough voting in Sections E-G and parameter studies in Section H. We also provide more qualitative results in Sections I and J, and analysis with error bars in Section K. At the end, we provide more discussions of the related works in Sections L, and broader impacts in Section M.

## B   DATASET DETAILS

We benchmark our proposed HoughST extensively over 11 widely adopted image recognition datasets. As Table 9 shows, the 11 datasets have rich diversity, spanning multi-style datasets with object images captured from several styles (e.g., real-world, synthetic, art, product and clipart styles) to task-specific datasets with real-world images for some specific visual task (e.g., the recognition of common objects, indoor and outdoor scenes, foods, traffic signs, natural textures and human actions). Below please find the detail of each dataset.

**Office** Saenko et al. (2010) includes 31-class images collected from Amazon (A), Webcam (W) and DSLR (D) styles which have 2817, 795 and 498 images, respectively. In addition to the original three styles in Office dataset, we further include an office Synthetic (S) style for benchmarking our HoughST comprehensively. The Synthetic (S) style is provided by Ringwald & Stiefelhagen (2021) and consists of 3100 images.

**Office-home** Venkateswara et al. (2017) consists of 65-class images collected from Art (A), Clipart (C), Product (P) and Real-World (R) styles which include 2496, 4464, 4503 and 4450 images, respectively.

**Adaptiope** Ringwald & Stiefelhagen (2021) has 123-class images collected from 3 styles, i.e., Product (P), Real-World (R) and Synthetic (S), where each style has 12300 images.

**VisDA** Peng et al. (2017) has over 280K images of 12 classes from Synthetic (S) style and Real-World (R) style, which contain 152397 and 127760 images, respectively.

**DomainNet** Peng et al. (2019) includes 345-class images from Clipart, Infograph, Painting, Quick-Draw, Real-World and Sketch styles which include 48129, 51605, 72266, 172500, 172947 and 69128 images, respectively.

**ImageNet** Deng et al. (2009) includes about 1.2M images that are uniformly distributed across the one thousand categories. The category annotation of ImageNet follows WordNet hierarchy and every image is annotated with one category label.

**SUN397** Xiao et al. (2010) has been proposed for scene recognition, which contains 39700 images covering 397 well-sampled scene categories, including indoor scenes and outdoor scenes.

**Food101** Bossard et al. (2014) is a real-world food dish dataset for fine-grained image recognition. The dataset consists of 101K images that cover 101 classes. Specifically, each class includes 250 cleaned test images and 750 purposely uncleaned training images.

**GTSRB** Stallkamp et al. (2011) is a real-world dataset for traffic signs recognition, which includes 50K images collected from various street scenes in Germany. These images have been labelled into 43 categories, including a training subset with 39209 images and a testing subset with 12630 images.

**Describable Textures (DTD)** Cimpoi et al. (2014) is a collection of textural images for texture recognition. This dataset consists of 5640 images with 47 categories, which have been uniformly separated into training, validation, and test subsets, where each subset contains 40 images per class. For each image, a main category and a list of the joint attributes are provided.

**UCF101** Soomro et al. (2012) has been proposed for benchmarking human action recognition with videos. It includes about 13K video clips of 101 actions, which are collected from YouTube. The video clips in the dataset have a resolution of 320x240 pixels and a frame rate of 25 FPS.

Table 9: Image recognition datasets used for vision-language model adaptation benchmark.

| Dataset | Classes | Images | Styles | Description |
|---|---|---|---|---|
| Office Saenko et al. (2010) | 31 | 4,110 | 4 | Office objects from Amazon, DSLR, Webcam and Synthetic styles. |
| Office-home Venkateswara et al. (2017) | 65 | 15,588 | 4 | Office and Home objects from Art, Clipart, Product and Real-World styles. |
| Adaptiope Ringwald & Stiefelhagen (2021) | 123 | 36,900 | 3 | Class-balanced object dataset with Product, Real-Life and Synthetic styles. |
| VisDA Peng et al. (2017) | 12 | 207,785 | 2 | A large-scale common object dataset with synthetic and real styles. |
| DomainNet Peng et al. (2019) | 345 | 586,575 | 6 | Common objects from Clipart, Infograph, Painting, Quickdraw, Real and Sketch styles. |
| ImageNet Deng et al. (2009) | 1,000 | 1,281,167 | 1 | A large-scale real-world object dataset with a wide range of categories. |
| SUN397 Xiao et al. (2010) | 397 | 76,129 | 1 | A real-world indoor and outdoor scenes dataset for scene understanding. |
| Food101 Bossard et al. (2014) | 101 | 75,750 | 1 | A real-world food dish dataset for food recognition. |
| GTSRB Stallkamp et al. (2011) | 43 | 26,640 | 1 | A real-world german traffic sign dataset for sign recognition. |
| DTD Cimpoi et al. (2014) | 47 | 3,760 | 1 | A real-world describable texture image dataset for texture perception. |
| UCF101 Soomro et al. (2012) | 101 | 9,537 | 1 | A real-world human action video dataset for action recognition. |

## C EXPERIMENTS WITH DIFFERENT BACKBONES

In the main manuscript, we study the generalization of our proposed HoughST by assessing it with three popular image recognition backbones, including two CNNs (i.e., ResNet-50 and ResNet-101) and one Transformer (i.e., ViT-B/16). Table 1 in the main manuscript provides the full results of the three backbones on multi-styles datasets Office, Office-Home and Adaptiope. Due to the space limit, Tables 2-4 the main manuscript only provide partial results for VisDA, DomainNet and other 5 task-specific datasets.

Here we provide the full result versions of the Tables 2-4 in the main manuscript, as shown in Tables 10-12, which further demonstrate that our HoughST works effectively and consistently over different image recognition backbones.

Table 10: VLMA performance (with three widely adopted backbone networks) on large-scale multi-style dataset VisDA. For fair comparisons, the results of all methods are based on the baseline CLIP.

| ViT-B/16 | VisDA Synthesis Style | | | | | | | | | | | |
|---|---|---|---|---|---|---|---|---|---|---|---|---|
| | plane | bcycl | bus | car | horse | knife | mcycl | person | plant | sktbrd | train | truck | Per-class |
| CLIP (baseline) | 98.5 | 99.7 | 64.6 | 92.5 | 99.7 | 96.8 | 85.3 | 98.4 | 99.8 | 79.4 | 66.4 | 73.4 | 87.8 |
| ST | 97.2 | 99.9 | 60.4 | 84.5 | 99.8 | 98.6 | 92.5 | 99.7 | 99.9 | 79.3 | 74.2 | 84.4 | 89.2 |
| CBST Zou et al. (2018) | 98.4 | 99.7 | 67.3 | 85.2 | 99.8 | 99.1 | 95.3 | 99.9 | 99.4 | 83.4 | 83.4 | 87.4 | 91.5 |
| CRST Zou et al. (2019) | 98.1 | 98.2 | 70.5 | 86.5 | 98.6 | 98.7 | 94.3 | 98.8 | 97.8 | 86.7 | 88.7 | 86.1 | 91.9 |
| SHOT Liang et al. (2020) | 99.6 | 99.1 | 74.6 | 86.3 | 98.3 | 99.3 | 96.4 | 96.1 | 99.7 | 87.5 | 90.1 | 87.3 | 92.2 |
| MUST Li et al. (2022a) | 98.7 | 99.2 | 76.3 | 86.4 | 99.6 | 99.2 | 95.3 | 99.3 | 99.8 | 89.2 | 89.9 | 82.6 | 92.9 |
| HoughST (Ours) | 99.7 | 99.7 | 78.9 | 86.6 | 99.9 | 99.3 | 96.4 | 99.4 | 99.8 | 91.9 | 90.8 | 93.2 | 94.6 |

| ViT-B/16 | VisDA Real Style | | | | | | | | | | | |
|---|---|---|---|---|---|---|---|---|---|---|---|---|---|
| | plane | bcycl | bus | car | horse | knife | mcycl | person | plant | sktbrd | train | truck | Per-class |
| CLIP (baseline) | 98.9 | 91.0 | 90.5 | 65.7 | 98.6 | 89.1 | 95.3 | 56.5 | 90.2 | 96.8 | 93.8 | 75.8 | 86.8 |
| ST | 99.4 | 87.3 | 92.5 | 68.3 | 98.1 | 90.4 | 94.6 | 69.3 | 91.2 | 96.7 | 94.5 | 66.4 | 87.3 |
| CBST Zou et al. (2018) | 99.3 | 89.2 | 91.3 | 76.9 | 98.2 | 89.5 | 95.4 | 68.1 | 88.4 | 96.4 | 94.1 | 64.2 | 87.5 |
| CRST Zou et al. (2019) | 99.1 | 90.7 | 91.4 | 64.5 | 99.1 | 93.4 | 95.1 | 68.2 | 91.3 | 96.8 | 95.3 | 67.2 | 87.6 |
| SHOT Liang et al. (2020) | 99.3 | 92.8 | 91.9 | 65.3 | 98.7 | 95.2 | 94.5 | 62.1 | 92.1 | 96.9 | 95.4 | 67.9 | 88.1 |
| MUST Li et al. (2022a) | 99.2 | 95.7 | 92.6 | 56.9 | 99.1 | 98.6 | 96.0 | 67.0 | 93.5 | 98.8 | 96.9 | 68.1 | 88.5 |
| HoughST (Ours) | 99.2 | 95.9 | 92.1 | 66.1 | 99.2 | 97.8 | 96.7 | 70.8 | 92.7 | 98.4 | 96.2 | 74.6 | 90.0 |

| ResNet-50 | VisDA Synthesis Style | | | | | | | | | | | |
|---|---|---|---|---|---|---|---|---|---|---|---|---|---|
| | plane | bcycl | bus | car | horse | knife | mcycl | person | plant | sktbrd | train | truck | Per-class |
| CLIP (baseline) | 96.0 | 99.1 | 43.4 | 92.4 | 98.5 | 94.5 | 69.6 | 92.1 | 99.1 | 46.6 | 53.0 | 41.5 | 77.1 |
| ST | 94.2 | 99.3 | 38.9 | 75.2 | 97.4 | 93.7 | 78.5 | 94.6 | 99.3 | 63.4 | 57.8 | 88.2 | 81.7 |
| CBST Zou et al. (2018) | 95.7 | 99.6 | 37.2 | 73.3 | 98.6 | 95.6 | 84.5 | 96.8 | 99.2 | 68.7 | 59.2 | 89.4 | 83.1 |
| CRST Zou et al. (2019) | 96.6 | 99.9 | 30.1 | 71.3 | 99.9 | 99.1 | 92.8 | 99.9 | 99.4 | 75.0 | 61.1 | 97.2 | 85.1 |
| SHOT Liang et al. (2020) | 97.3 | 99.9 | 43.7 | 73.4 | 98.6 | 98.6 | 86.5 | 94.5 | 99.1 | 77.3 | 68.9 | 84.4 | 86.0 |
| HoughST (Ours) | 97.6 | 99.8 | 57.2 | 84.7 | 99.9 | 98.7 | 91.7 | 99.8 | 100 | 79.2 | 74.5 | 83.1 | 88.8 |

| ResNet-50 | VisDA Real Style | | | | | | | | | | | |
|---|---|---|---|---|---|---|---|---|---|---|---|---|---|
| | plane | bcycl | bus | car | horse | knife | mcycl | person | plant | sktbrd | train | truck | Per-class |
| CLIP (baseline) | 97.3 | 82.1 | 83.0 | 55.4 | 96.7 | 73.4 | 91.1 | 59.9 | 86.6 | 93.4 | 91.8 | 73.8 | 82.0 |
| ST | 97.6 | 78.1 | 99.7 | 65.9 | 96.2 | 79.3 | 90.1 | 62.8 | 82.9 | 94.2 | 94.1 | 74.3 | 84.1 |
| CBST Zou et al. (2018) | 95.8 | 83.2 | 80.3 | 54.5 | 96.8 | 92.2 | 92.1 | 78.8 | 91.6 | 88.8 | 89.8 | 76.0 | 84.9 |
| CRST Zou et al. (2019) | 96.9 | 86.9 | 83.1 | 71.1 | 93.4 | 91.9 | 91.7 | 80.3 | 90.2 | 89.4 | 88.5 | 65.6 | 85.7 |
| SHOT Liang et al. (2020) | 96.5 | 85.4 | 85.4 | 59.6 | 96.3 | 94.8 | 92.7 | 80.3 | 92.4 | 90.5 | 90.4 | 75.4 | 86.6 |
| HoughST (Ours) | 97.2 | 87.2 | 88.2 | 78.1 | 97.2 | 95.1 | 93.0 | 81.5 | 92.1 | 91.2 | 92.7 | 65.6 | 88.2 |

| ResNet-101 | VisDA Synthesis Style | | | | | | | | | | | |
|---|---|---|---|---|---|---|---|---|---|---|---|---|---|
| | plane | bcycl | bus | car | horse | knife | mcycl | person | plant | sktbrd | train | truck | Per-class |
| CLIP (baseline) | 96.8 | 99.4 | 24.2 | 87.5 | 98.9 | 96.7 | 83.1 | 58.2 | 99.3 | 61.2 | 47.1 | 72.4 | 77.0 |
| ST | 95.2 | 99.6 | 26.7 | 84.3 | 99.1 | 97.2 | 84.2 | 91.3 | 99.5 | 68.4 | 57.6 | 81.2 | 82.0 |
| CBST Zou et al. (2018) | 96.7 | 99.8 | 27.3 | 74.5 | 99.9 | 99.5 | 93.8 | 99.9 | 100 | 73.1 | 62.3 | 97.0 | 85.3 |
| CRST Zou et al. (2019) | 96.9 | 99.9 | 42.0 | 78.6 | 99.9 | 98.9 | 93.5 | 99.9 | 99.9 | 73.0 | 72.0 | 94.4 | 87.4 |
| SHOT Liang et al. (2020) | 98.5 | 99.7 | 39.9 | 83.1 | 100 | 98.5 | 97.8 | 96.1 | 100 | 79.3 | 81.7 | 91.3 | 89.0 |
| HoughST (Ours) | 97.8 | 99.8 | 47.5 | 85.5 | 100 | 98.8 | 96.6 | 99.9 | 100 | 81.1 | 83.2 | 92.2 | 90.2 |

| ResNet-101 | VisDA Real Style | | | | | | | | | | | |
|---|---|---|---|---|---|---|---|---|---|---|---|---|---|
| | plane | bcycl | bus | car | horse | knife | mcycl | person | plant | sktbrd | train | truck | Per-class |
| CLIP (baseline) | 97.8 | 83.7 | 87.9 | 76.2 | 97.4 | 77.9 | 93.8 | 53.7 | 84.3 | 90.7 | 91.0 | 67.2 | 83.4 |
| ST | 97.4 | 84.7 | 86.6 | 75.2 | 97.1 | 80.5 | 94.1 | 69.6 | 89.6 | 91.1 | 92.3 | 68.7 | 85.5 |
| CBST Zou et al. (2018) | 97.3 | 86.5 | 87.7 | 70.6 | 97.3 | 93.8 | 93.3 | 74.5 | 91.7 | 89.1 | 91.5 | 69.1 | 86.8 |
| CRST Zou et al. (2019) | 97.5 | 82.9 | 86.3 | 82.2 | 97.8 | 93.1 | 95.4 | 68.5 | 91.3 | 91.3 | 93.2 | 66.8 | 87.4 |
| SHOT Liang et al. (2020) | 97.3 | 88.6 | 88.6 | 69.8 | 97.3 | 94.2 | 92.9 | 80.4 | 91.8 | 92.7 | 92.3 | 69.2 | 87.9 |
| HoughST (Ours) | 97.8 | 89.1 | 88.3 | 78.3 | 97.3 | 94.5 | 94.7 | 82.1 | 92.8 | 93.6 | 93.8 | 69.5 | 89.3 |

Table 11: VLMA performance (with three widely adopted backbone networks) on task-specific datasets of various image recognition tasks. For fair comparisons, the results of all methods are based on the baseline CLIP.

| Method | ViT-B/16 | | | | | | ResNet-50 | | | | | |
|---|---|---|---|---|---|---|---|---|---|---|---|---|
| | SUN397 | Food101 | GTSRB | DTD | UCF101 | Mean | SUN397 | Food101 | GTSRB | DTD | UCF101 | Mean |
| CLIP (baseline) | 60.8 | 85.6 | 32.5 | 44.5 | 64.1 | 57.5 | 54.0 | 73.1 | 25.0 | 39.8 | 56.0 | 49.5 |
| ST | 65.8 | 88.2 | 32.8 | 45.0 | 67.0 | 59.7 | 59.0 | 74.4 | 20.5 | 35.8 | 56.4 | 49.2 |
| CBST Zou et al. (2018) | 63.2 | 89.5 | 37.6 | 44.3 | 68.1 | 60.5 | 63.7 | 78.2 | 27.4 | 38.7 | 59.5 | 53.5 |
| CRST Zou et al. (2019) | 64.7 | 89.1 | 39.7 | 45.3 | 68.6 | 61.4 | 64.2 | 76.5 | 30.1 | 39.4 | 61.3 | 54.3 |
| SHOT Liang et al. (2020) | 66.1 | 89.6 | 41.2 | 46.3 | 69.4 | 62.5 | 65.1 | 77.3 | 34.6 | 41.2 | 62.7 | 56.1 |
| MUST Li et al. (2022a) | 67.7 | 89.4 | 42.7 | 46.5 | 70.6 | 63.3 | - | - | - | - | - | - |
| HoughST (Ours) | 71.8 | 91.1 | 49.3 | 52.7 | 73.9 | 67.7 | 65.7 | 79.5 | 39.6 | 49.4 | 65.6 | 59.9 |

| Method | ResNet-101 | | | | | |
|---|---|---|---|---|---|---|
| | SUN397 | Food101 | GTSRB | DTD | UCF101 | Mean |
| CLIP (baseline) | 51.5 | 82.3 | 27.5 | 37.8 | 58.3 | 51.4 |
| ST | 56.5 | 79.9 | 23.6 | 35.4 | 60.2 | 51.1 |
| CBST Zou et al. (2018) | 65.7 | 81.5 | 28.3 | 37.3 | 60.5 | 54.6 |
| CRST Zou et al. (2019) | 61.4 | 80.7 | 31.4 | 37.3 | 63.0 | 54.7 |
| SHOT Liang et al. (2020) | 63.7 | 81.4 | 33.9 | 42.5 | 64.3 | 57.1 |
| MUST Li et al. (2022a) | - | - | - | - | - | - |
| HoughST (Ours) | 67.5 | 83.4 | 38.2 | 48.1 | 66.2 | 60.6 |

# D    PSEUDO CODES OF HOUGH VOTING-BASED SELF-TRAINING

We provide the pseudo codes of our proposed Hough voting-based self-training (HoughST), as shown in Algorithm 1. Note Algorithm 1 describes the unsupervised adaptation process in a epoch-

Table 12: VLMA performance (with three widely adopted backbone networks) on multi-style datasets of DomainNet. For fair comparisons, the results of all methods are based on the baseline CLIP.

| Method | ViT-B/16 | | | | | | | ResNet-50 | | | | | | |
|---|---|---|---|---|---|---|---|---|---|---|---|---|---|---|
| | Clipart | Info | Paint | Quick | Real | Sketch | Mean | Clipart | Info | Paint | Quick | Real | Sketch | Mean |
| CLIP (baseline) | 69.7 | 47.8 | 65.0 | 14.5 | 82.0 | 62.4 | 56.9 | 51.9 | 39.1 | 52.1 | 6.4 | 74.7 | 47.4 | 45.3 |
| ST | 72.5 | 51.3 | 68.7 | 12.4 | 83.7 | 64.3 | 58.8 | 55.4 | 40.5 | 54.8 | 4.3 | 76.2 | 48.3 | 46.5 |
| CBST Zou et al. (2018) | 74.3 | 56.8 | 69.8 | 13.4 | 83.1 | 67.1 | 60.7 | 56.3 | 40.7 | 56.2 | 5.6 | 77.4 | 48.1 | 47.3 |
| CRST Zou et al. (2019) | 75.6 | 56.9 | 71.3 | 14.8 | 83.3 | 68.2 | 61.7 | 57.9 | 41.8 | 57.1 | 6.2 | 78.2 | 49.5 | 48.4 |
| SHOT Liang et al. (2020) | 75.9 | 57.4 | 71.5 | 15.1 | 83.3 | 68.8 | 62.0 | 60.3 | 45.8 | 60.5 | 5.1 | 78.9 | 54.1 | 50.8 |
| MUST Li et al. (2022a) | 76.1 | 57.5 | 71.6 | 14.2 | 84.4 | 68.9 | 62.1 | - | - | - | - | - | - | - |
| HoughST (Ours) | 77.6 | 59.0 | 73.1 | 18.2 | 86.1 | 70.1 | 64.0 | 62.7 | 47.2 | 61.3 | 7.2 | 80.2 | 54.4 | 52.2 |

| Method | ResNet-101 | | | | | | |
|---|---|---|---|---|---|---|---|
| | Clipart | Info | Paint | Quick | Real | Sketch | Mean |
| CLIP (baseline) | 58.8 | 41.5 | 58.0 | 8.9 | 77.4 | 53.8 | 49.8 |
| ST | 61.4 | 47.5 | 61.7 | 6.1 | 78.9 | 55.2 | 51.8 |
| CBST Zou et al. (2018) | 63.2 | 48.3 | 62.5 | 6.7 | 79.4 | 56.1 | 52.7 |
| CRST Zou et al. (2019) | 64.3 | 49.4 | 63.2 | 6.9 | 80.2 | 57.8 | 53.6 |
| SHOT Liang et al. (2020) | 66.4 | 49.4 | 65.4 | 7.9 | 80.8 | 59.2 | 54.9 |
| MUST Li et al. (2022a) | - | - | - | - | - | - | - |
| HoughST (Ours) | 69.6 | 50.8 | 65.9 | 9.5 | 82.5 | 60.4 | 56.4 |

wise manner for simple illustration and presentation. In experiments, we implement Algorithm 1 in a iteration-wise manner with mini-batches. Besides, Lines 7-8 in Algorithm 1 can be skipped in the first training iteration as the model has not been updated at that time.

Our HoughST introduces Hough voting into self-training, where the voting centroids and the model are alternatively updated as illustrated in Line 8 and Line 10 in Algorithm 1. In this way, HoughST captures temporal information via temporal Hough voting, which helps memorize previously learnt downstream dataset information via voting from the features encoded by the intermediate models evolved along the adaptation process.

---

**Algorithm 1** Hough Voting-based Self-training.

---

**Require:** Target images $X^I$, target class descriptions $X^T$ and a pre-trained vision-language model $F = \{F^I, F^T\}$
**Ensure:** Adapted vision-language model $F$
1: **Initialization:**
2: Calculate textual Hough voting centroid $\delta_m^T$ using $X^T$ and $F$ via Eq. 4
3: Calculate visual Hough voting centroid $\delta_m^I$ using $X^I$ and $F$ via Eq. 5
4: Initialize vision-language Hough voting centroid $\delta_m^{IT}$ using $\delta_m^T$ and $\delta_m^I$ as in the left part of Eq. 6
5: **for** $epoch = 1$ **to** $Max\_Epoch$ **do**
6:     **Pseudo Label Generation:**
7:     Calculate new visual Hough voting centroid $\delta_m^I$ using $X^I$ and the updated $F$ using Eq. 5
8:     Update vision-language Hough voting centroid $\delta_m^{IT}$ with new visual Hough voting centroid $\delta_m^I$ as in the right part of Eq. 6
9:     Generate pseudo labels $Y^I$ with the updated vision-language Hough voting centroid $\delta_m^{IT}$ via Eq. 7
10:     **Network Optimization with Pseudo Labels:**
11:     Optimize $F$ using pseudo labels $Y^I$ via Eq. 8
12: **end for**
13: **return** $F$

---

# E HOW LLM-GENERATED TEXT DESCRIPTIONS AFFECT OTHER METHODS

As described in Section 3, our proposed HoughST adopts GPT-3 Brown et al. (2020) as the large language model (LLM) to generate multiple text descriptions for a given class for mitigating distribution shifts in text modality. For comprehensively benchmarking HoughST, we provide the results of the state-of-the-art methods using the same LLM-generated text descriptions as those used in HoughST. Table 13 presents the results on dataset Office with backbone ViT-B/16. We can observe that directly using LLM-generated text descriptions for these methods improves the performance slightly. Beside, it can be seen that our HoughST still outperforms the state-of-the-arts that used

LLM-generated text descriptions, largely because HoughST conducts Hough voting-based learning that filters out noisy textual information, fuses and updates the textual information, and utilize them to denoise pseudo labels.

Table 13: Results of the state-of-the-art methods with the text descriptions generated from Large Language Models Brown et al. (2020). For fair comparisons, the results of all methods are based on the baseline CLIP.

| ViT-B/16 | Office | | | | |
|---|---|---|---|---|---|
| | A | W | D | S | Mean |
| ST | 78.6 | 81.1 | 78.3 | 68.6 | 76.6 |
| ST + LLM Brown et al. (2020) | 79.2 | 82.0 | 78.9 | 70.1 | 77.5 |
| CBST Zou et al. (2018) | 79.1 | 80.7 | 78.5 | 68.9 | 76.8 |
| CBST Zou et al. (2018) + LLM Brown et al. (2020) | 80.1 | 81.4 | 79.3 | 70.3 | 77.7 |
| CRST Zou et al. (2019) | 78.8 | 81.2 | 79.1 | 69.0 | 77.0 |
| CRST Zou et al. (2019) + LLM Brown et al. (2020) | 79.1 | 82.1 | 80.3 | 70.2 | 77.9 |
| SHOT Liang et al. (2020) | 79.2 | 81.1 | 81.2 | 67.1 | 77.1 |
| SHOT Liang et al. (2020) + LLM Brown et al. (2020) | 80.7 | 81.9 | 81.7 | 68.9 | 78.3 |
| MUST Li et al. (2022a) | 79.0 | 81.4 | 79.5 | 69.2 | 77.2 |
| MUST Li et al. (2022a) + LLM Brown et al. (2020) | 81.2 | 82.1 | 80.7 | 70.2 | 78.5 |
| HoughST (Ours) | **84.3** | **82.8** | **81.3** | **72.3** | **80.1** |

## F  HOUGHST WITH DIFFERENT LLMS

As described in the main manuscript, our proposed HoughST employs GPT-3 Brown et al. (2020) as the large language model (LLM) to generate multiple text descriptions for a given class. Specifically, for all datasets, we query the large language model with the following input:

"Describe what a/an `[class name]`, a type of `[dataset name]`, looks like."

In this section, we study how the adoption of LLM affects HoughST by implementing HoughST with different LLMs, including GPT-3 Brown et al. (2020), GPT-2 Radford et al. (2019) and GPT-J-6B Wang & Komatsuzaki (2021). Experimental results in Table 14 show that the change of LLM does not affect HoughST clearly, demonstrating that HoughST can work effectively and consistently with different qualities of text descriptions (generated by different LLMs).

Table 14: HoughST with different large language models. Experiments are conducted with ViT-B/16 on dataset Office. The default implementation is highlighted in  gray .

| Method | Office (Mean) | Office-home (Mean) | Adaptiope (Mean) |
|---|---|---|---|
| ST | 76.6 | 75.4 | 72.7 |
| HoughST (GPT-2 Radford et al. (2019)) | 79.3 | 77.5 | 78.3 |
| HoughST (GPT-J-6B Wang & Komatsuzaki (2021)) | 79.2 | 77.9 | 78.8 |
| HoughST (GPT-3 Brown et al. (2020)) | 80.1 | 78.9 | 79.9 |

## G  MORE DISCUSSION OF TEXTUAL HOUGH VOTING

As described in the main manuscript, the proposed textual Hough voting fuses text features in a two-step manner: 1) uniformly average the multiple text features to acquire an initial voting centroid; 2) calculate the final voting centroid by weighted average where the weight of each feature is the distance between it and the initial voting centroid. This two-step voting operation allows smooth feature fusion by weighting down the effect of corner cases, which is important for textual Hough voting as the LLM-generated text descriptions are not always reliable (e.g., when experiencing generation failures, LLM may generate only a full stop character "." or a random word).

In this section, we conduct experiments with ViT-B/16 on ImageNet to investigate the effect of this two-step feature fusion strategy on our proposed Hough voting. Table 15 shows that the two-step feature fusion strategy brings about 0.4% performance improvement on ImageNet, largely because it allows smooth feature fusion by down-weighting the effect of corner cases.

Table 15: Textual Hough Voting (THV) with and without the two-step feature fusion strategy. Experiments are conducted with ViT-B/16 on ImageNet. The default implementation is highlighted in gray .

| Method | ImageNet |
|---|---|
| CLIP (baseline) | 68.3 |
| THV (w/o two-step feature fusion strategy) | 69.4 |
| THV (w/ two-step feature fusion strategy) | 69.8 |

## H    MORE PARAMETER STUDIES

As described in the main manuscript, our proposed HoughST employs the large language model to generate $K$ text descriptions for each class for achieving textual Hough voting. We investigate $K$ by varying it from 10 to 25, as shown in Table 16. It can be seen that varying $K$ does not affect the proposed HoughST clearly, demonstrating that our HoughST is quite tolerant to the hyper-parameter $K$.

Table 16: Parameter study for the number of text descriptions $K$ with ViT-B/16 on Office. The default value is marked in gray .

| Parameter $K$ | 10 | 15 | 20 | 25 |
|---|---|---|---|---|
| Office (Mean) | 79.9 | 80.1 | 80.1 | 80.0 |

As described in the main manuscript, our proposed HoughST introduces visual Hough voting that employs the off-the-shelf image augmentation policies in Cubuk et al. (2020) to generate $K$ augmentations for all images respectively, which are then selectively fused using pseudo class labels to describe each class. We investigate $K$ by varying it from 10 to 25, as shown in Table 17. It can be seen that varying $K$ does not affect the proposed HoughST clearly, demonstrating that our HoughST is quite tolerant to the hyper-parameter $K$.

Table 17: Parameter study for the number of augmented image data $K$ with ViT-B/16 on dataset Office. The default value is marked in gray .

| Parameter $K$ | 10 | 15 | 20 | 25 |
|---|---|---|---|---|
| Office (Mean) | 80.0 | 80.1 | 80.1 | 79.9 |

## I    MORE PSEUDO LABEL ACCURACY FIGURES

In the main manuscript, we provide the pseudo label accuracy along the unsupervised adaptation process for Office datasets.

In this section, we provide the pseudo label accuracy figures over more datasets, i.e., Office-home, Adaptiope, VisDA, SUN397, Food101, GTSRB, DTD, UCF101, and ImageNet. Fig. 5 shows the pseudo label accuracy along the unsupervised adaptation process with the backbone ViT-B/16. It can be seen that our proposed HoughST generates much more accurate pseudo labels than the vanilla self-training (ST) and the state-of-the-art MUST consistently over various datasets. The superior pseudo label accuracy is largely attributed to the proposed Hough voting-based self-training which helps capture rich target image and text information that is more invariant to visual and textual distribution shifts and can lead to better unsupervised self-training.

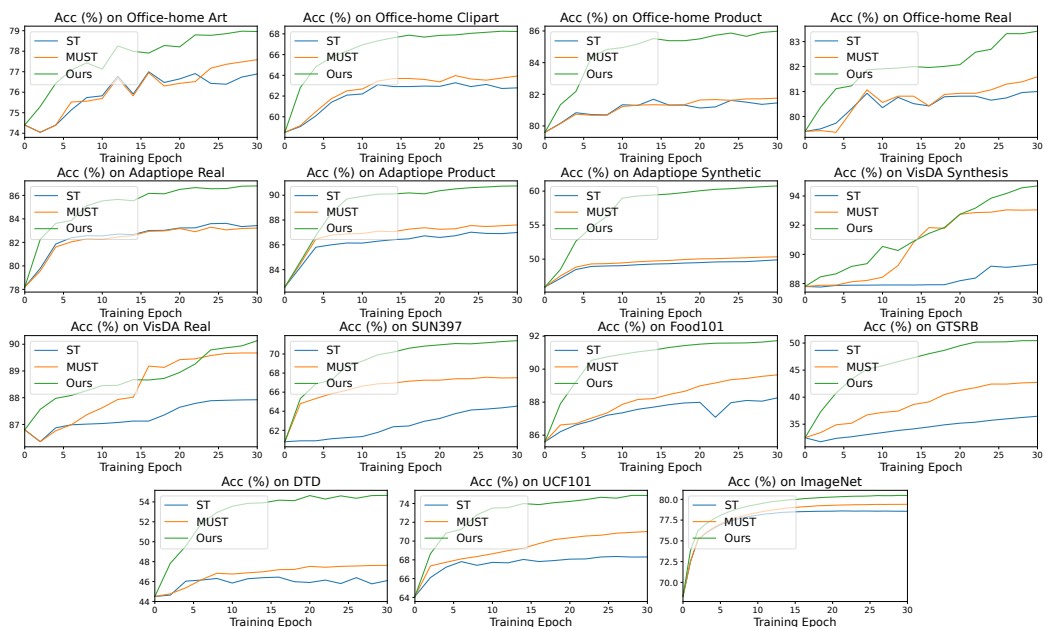

Figure 5: Pseudo label accuracy along the unsupervised adaptation process in VLMA: The experiments were conducted over 10 widely adopted datasets and all use ViT-B/16. The results on dataset Office are provided in the main manuscript.

## J  QUALITATIVE RESULTS

We illustrate our proposed HoughST qualitatively by providing class activation map Selvaraju et al. (2017) (CAM) visualization on dataset Office with ViT-B/16. Fig. 6 provides the CAMs of ST (2nd column), MUST Li et al. (2022a) (3rd column) and our HoughST (4th column). We can observe that our proposed HoughST preforms image recognition based on more diverse image regions, leading to robust and accurate visual recognition under large distribution shifts. For example, in the recognition of backpack, HoughST tends to rely on more image regions (e.g., various local regions with zippers) which together form a holistic representation of this backpack, ultimately leading to a robust prediction under large distribution shifts. As a comparison, ST and MUST Li et al. (2022a) make predictions largely according to a single image region and pay less attentions on other image regions, which may lead to performance degradation when experiencing large distribution shifts. The CAMs of Mountain Bike and Helmet shown in the second and third rows respectively are consistent with the above observation.

## K  ANALYSIS WITH ERROR BARS

In experiments, we observe negligible variance on the results between multiple random runs. Nevertheless, we provide the error bar with 5 random runs to analyze the proposed HoughST with ViT-B/16 on Office dataset, as shown in Table 18. It shows that our proposed HoughST performs well consistently over multiple random runs.

Table 18: Analysis of our proposed HoughST with error bars. Experiments are conducted with ViT-B/16.

| Method | Office (Mean) | Office-home (Mean) | Adaptiope (Mean) |
|---|---|---|---|
| HoughST | 80.1 ± 0.1 | 78.9 ± 0.1 | 79.9 ± 0.2 |

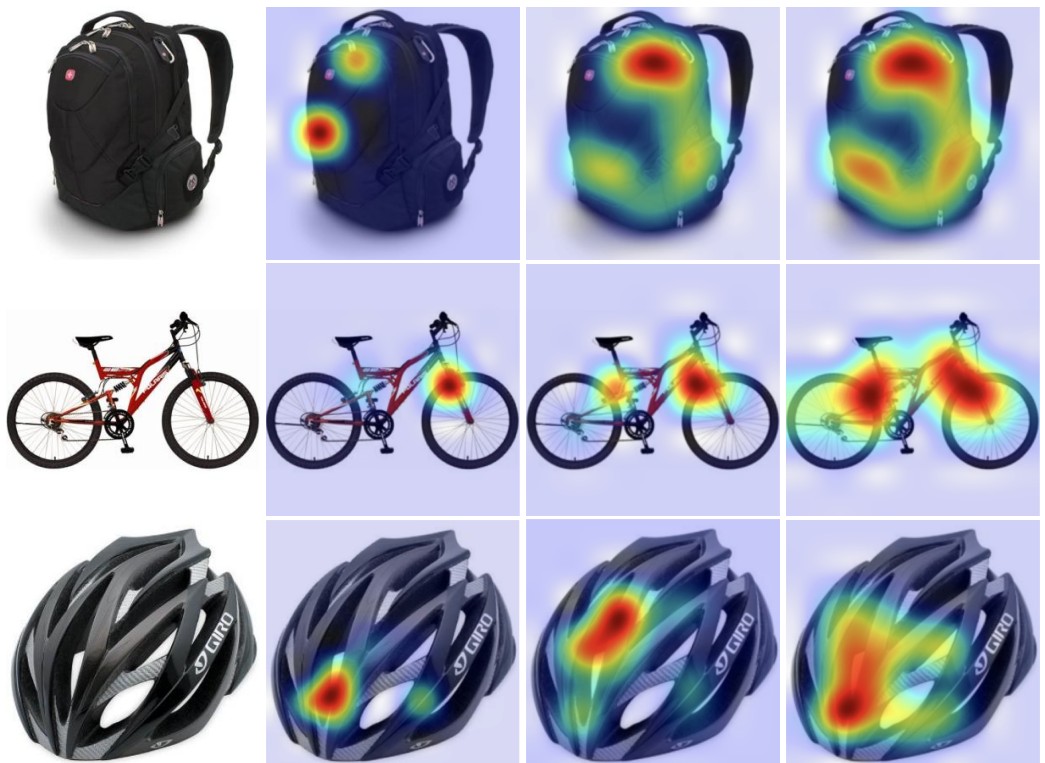

Figure 6: **Qualitative comparisons** with class activation maps Selvaraju et al. (2017) (CAM) on dataset Office with ViT-B/16. The 4 columns from left to right show Input Images and the corresponding CAMs by ST, MUST Li et al. (2022a) and our HoughST, respectively. It can be observed that HoughST preforms image recognition based on more diverse image regions, leading to more robust and accurate visual recognition under various cross-dataset scenarios.

## L  RELATIONS TO OPEN-SET, CLASS-INCREMENTAL AND PARTIAL DOMAIN ADAPTATION

Different from traditional domain adaptation that assumes the same vocabulary across source and downstream datasets, this work studies vision-language model adaptation (VLMA), a new unsupervised model adaptation (UMA) framework that positions a pre-trained VLM as the source model and transfers it towards various unlabelled downstream datasets.

We note that there are several other adaptation frameworks which also aim to handle the situation where the pre-training and downstream datasets have different vocabularies. In this section, we briefly introduce their frameworks and clarify the difference between them and the studied VLMA.

Specifically, open-set domain adaptation Panareda Busto & Gall (2017); Saito et al. (2018); Liu et al. (2019), class-incremental domain adaptation Kundu et al. (2020); Xu et al. (2021) and partial domain adaptation Cao et al. (2018; 2019); Zhang et al. (2018), are proposed to handle the situation where the source and downstream datasets have different vocabularies. However, all these frameworks have certain limitations as compared the studied VLMA.

For example, **open-set domain adaptation** Panareda Busto & Gall (2017); Saito et al. (2018); Liu et al. (2019) adds an extra class called "unknown" to both source and downstream datasets such that it allows open-set adaptation by treating all the classes that are not shared between source and downstream datasets as the "unknown" class. However, open-set domain adaptation can merely classify all new target classes/concepts as a single "unknown" class even in an ideal case, which fails to respectively recognize new classes/concepts, limiting its flexibility and efficiency greatly in unsupervised transfer. Differently, VLMA allows to respectively recognize various new downstream categories/concepts, which is much more flexible.

**Class-incremental domain adaptation** Kundu et al. (2020); Xu et al. (2021) integrates domain adaptation and class-incremental learning (using one-shot or few-shot labelled downstream images) such that it allows to recognize new target classes/concepts during adaptation. However, it generally requires one-shot or few-shot labelled downstream images for each new class as a prerequisite, while VLMA is unsupervised and can work for new classes without requiring labelled target images.

**Partial domain adaptation** Cao et al. (2018; 2019); Zhang et al. (2018) assumes that the label set of downstream dataset is a subset of the label set of source dataset. Differently, the studied VLMA does not have this constraint as it can work with various downstream classes Radford et al. (2015).

## M  BROADER IMPACTS AND LIMITATIONS

We envision that this work will promote more studies on VLMA, a new unsupervised model adaptation framework that mitigates the image annotation constraint and facilitate deep network training while handling new visual recognition tasks. Furthermore, as our work is built upon open-source pre-trained vision-language models, it adds only a small amount of computation overhead after VLM pre-training and therefore reduces the carbon footprint. Currently, we do not foresee clear undesirable impacts of this work from both ethical and social aspects. At the other hand, the investigated techniques in this work are still at a very early stage and thus the proposed approach could be used as an assistant tool in computer vision applications instead of the critical decision and hard control systems that may lead to severe and harmful consequences.

