# OpenReview forum: "Hough Voting-based Self-Training for Vision-Language Model Adaptation"
_ICLR.cc/2025/Conference — ICLR 2025 Conference Withdrawn Submission_

### Official Review · Reviewer_2Z4t · 2024-10-27

**Soundness:** 3
**Presentation:** 3
**Contribution:** 2
**Rating:** 5
**Confidence:** 5

**Summary:**

This paper introduces Hough Voting-based Self-Training (HoughST), a novel framework for vision-language model adaptation (VLMA). HoughST leverages a multimodal Hough voting mechanism to align visual and textual features, mitigating distribution shifts in both modalities. It also incorporates temporal Hough voting to capture and utilize previously learned information, enhancing the self-training process. Extensive experiments across 11 image recognition tasks demonstrate its effectiveness in unsupervised model adaptation.

**Strengths:**

1.	Clarity and Logical Structure. The paper is well-structured and easy to follow. Each section is clearly organized, and the flow of information is logical and coherent, making it accessible and engaging for readers.
2.	Originality and Innovation. The paper introduces a novel approach by integrating Hough Voting into the self-training process for vision-language model adaptation (VLMA). The multimodal Hough voting mechanism, covering visual, textual, and temporal dimensions, is a creative and effective solution to address distribution shifts in both modalities.
3.	Fairness of Comparative Evaluation. The authors use multiple datasets and different backbone architectures to validate HoughST's effectiveness. All methods are based on the same baseline (CLIP), and the results are presented in a transparent and consistent manner, providing a reliable basis for comparison.

**Weaknesses:**

1. Lack of Clarity in Method Figures. The method structure is unclear. Figure 2 does not show the process of updating the Multimodal Codebook or indicate that only the visual feature extractor is updated. The core component, Multimodal Hough Voting, lacks a detailed structural description, making it difficult to understand the method's process.

2. Unclear Formula Definitions. The formulas are not clearly defined. The symbol “1(y ̂_n^I==m) ” in formula (3) is not explained, leading to potential confusion for readers. Clear definitions of all symbols are necessary for the paper to be comprehensible and reproducible.

3. Insufficient Experimental Comparison.  The experimental comparison is insufficient. The latest method used for comparison is from 2022, and there is no comparison with current state-of-the-art (SOTA) methods[1][2][3]. Including more recent and advanced methods would provide a more robust evaluation of the proposed approach.

[1]Zhang, S., Naseer, M., Chen, G., Shen, Z., Khan, S., Zhang, K., & Shahbaz Khan, F. (2024). S3A: Towards Realistic Zero-Shot Classification via Self Structural Semantic Alignment. Proceedings of the AAAI Conference on Artificial Intelligence, 38(7), 7278-7286. https://doi.org/10.1609/aaai.v38i7.28557

[2]Qiu, L., Zhang, Q., Chen, X., & Cai, S. (2024). Multi-Level Cross-Modal Alignment for Image Clustering. Proceedings of the AAAI Conference on Artificial Intelligence, 38(13), 14695-14703. https://doi.org/10.1609/aaai.v38i13.29387

[3]Han, K., Li, Y., Vaze, S., Li, J., & Jia, X. (2023). What's in a Name? Beyond Class Indices for Image Recognition. arxiv preprint arxiv:2304.02364.

**Questions:**

1.	Why were more recent state-of-the-art methods not included in the experimental comparison? Could you consider adding these methods to provide a more comprehensive evaluation?
2.	Why are the textual and visual Hough voting centroids directly added together rather than using a weighted average in formula (6)? Could you provide a rationale for this choice and discuss any potential benefits or drawbacks of using a weighted average instead?
3.	Consider adding annotations or labels to Figure 2 to highlight the key components and steps involved in the Multimodal Hough Voting process.

---

### Official Review · Reviewer_kUXo · 2024-10-28

**Soundness:** 2
**Presentation:** 2
**Contribution:** 2
**Rating:** 5
**Confidence:** 4

**Summary:**

The paper tackles the problem of unsupervised adaptation of vision-language models (e.g., CLIP) to downstream datasets that could present domain shifts. The core idea of the paper is Hough voting based self training scheme to counter the domains shifts in visual and textual spaces from source to target downstream datasets. The proposed multimodal hough voting mechanism considers textual hough voting, visual hough voting and temporal hough voting during the self-training process to improve the pseudo-labelling performance. Experiments have been performed on several datasets, including domain shifts and tasks-specific downstream datasets. Results claim to achieve improved performance over the existing methods and the baseline method.

**Strengths:**

**1**. The paper addresses an important problem of adapting vision language models to downstream datasets where there could be domain shifts.

**2**. Hough voting based self training takes into account different aspects of multimodal adaptation, such as image, textual features and temporal context into account to derive the pseudo-label for self-training.

**3**. The paper considers a variety of datasets to conduct experiments, including used in domain adaptation literature as well as vision-language model adaptation literature.

**4**. Results claim to surpass the existing methods and baselines consistently.

**Weaknesses:**

**1**. The novelty of the method seems weak as the all Hough voting based self-training is doing is primarily building textual and image prototypes and then using an EMA mechanism to update the so-called multimodal feature.

**2**. The experiments does not consider some recent and relevant methods for comparison e.g., [A], [B], and [C]. This makes the evaluation weaker and claimed effectiveness questionable.

**3**. The paper uses LLM generated descriptions for classes and then develop a prototype, that it calls textual Hough voting, however, the same ideas have been explored in previous papers such as [C], [D]. It seems that the baselines considered in the paper doesn’t use (same) LLM generated descriptions for a fair comparison.

**4**. The results of MUST with VIT-B/16 does not match with the ones reported in MUST paper for all task-specific datasets? Is there a specific reason for this?

**5**. It is not clear how the proposed idea helps in countering the domain shifts using the proposed Hough-based self training approach. Lack of such study, makes the grounding of the proposed approach is quite weak.

**6**. The ablation study is shown only one dataset (Office) and there are several other datasets used in the overall set of experiments.

[A] Mirza, M.J., Karlinsky, L., Lin, W., Possegger, H., Kozinski, M., Feris, R. and Bischof, H., 2024. Lafter: Label-free tuning of zero-shot classifier using language and unlabeled image collections. Advances in Neural Information Processing Systems, 36.

[B] Hu, X., Zhang, K., Xia, L., Chen, A., Luo, J., Sun, Y., Wang, K., Qiao, N., Zeng, X., Sun, M. and Kuo, C.H., 2024. Reclip: Refine contrastive language image pre-training with source free domain adaptation. In Proceedings of the IEEE/CVF Winter Conference on Applications of Computer Vision (pp. 2994-3003).

[C] Pratt, S., Covert, I., Liu, R. and Farhadi, A., 2023. What does a platypus look like? generating customized prompts for zero-shot image classification. In Proceedings of the IEEE/CVF International Conference on Computer Vision (pp. 15691-15701).

[D] Saha, O., Van Horn, G. and Maji, S., 2024. Improved Zero-Shot Classification by Adapting VLMs with Text Descriptions. In Proceedings of the IEEE/CVF Conference on Computer Vision and Pattern Recognition (pp. 17542-17552).

[E] Liang, K.J., Rangrej, S.B., Petrovic, V. and Hassner, T., 2022. Few-shot learning with noisy labels. In Proceedings of the IEEE/CVF Conference on Computer Vision and Pattern Recognition (pp. 9089-9098).

**Questions:**

**1**. It is essential to fairly compare the proposed Hough voting based self training with [A], [B] and [C]. I would like to see a clear comparison with such approaches on task-specific datasets and a couple of OOD datasets.

**2**. Is self-training based baseline also using same augmentations used in the proposed method?

**3**. Does the proposed mechanism also help towards convergence of the model? How the best model is selected? Is there any validation set used for each of the datasets?

**4**. I would like to see comparison with CLIP zero-shot and self-training baseline with the same LLM generated descriptions as used in the proposed method on datasets other than Office (Table 13 appendix).

**5**. What makes the proposed Hough based voting mechanism combat the various domain shifts in OfficeHome and DomainNet type datasets. Some analyses to establish such OOD robustness of the model is necessary to develop trust in the main claim of the paper.

**6**. I would like to see ablation study on DTD, Food 101 and GSTSRB datasets to better understand the contribution of each of the components.

**7**. The paper needs to cite some very relevant and recent methods (e.g., [A], [B]) trying to tackle the same problem of unsupervised adaptation of vision-language models.

**8**. I would like to see how the proposed method compares against a simple baseline of confidence-based thresholding mechanism for selecting pseudo-label.

**9**. It would be important to see an elegant explanation on how the proposed idea is different to typical methods that construct (weighted) prototypes to perform few-shot learning such as [E].

**10**. The (main) paper considers CLIP (zero-shot) as baseline to report significant improvements (L:369-371), however, it sounds a bit misleading as the actual baseline is (naive) self-training with LLM generated textual descriptions.

**11** .L482: How the proposed multimodal Hough voting captures target dataset information that is invariant to distribution shifts?

**12**. Fig. 6 (appendix): What makes the proposed method focus on more diverse regions? And why it is a better idea? Also, is it the case with other datasets such as DTD and Food101 as well?

---

### Official Review · Reviewer_yCxK · 2024-10-31

**Soundness:** 3
**Presentation:** 2
**Contribution:** 2
**Rating:** 3
**Confidence:** 4

**Summary:**

This paper studies vision-language model adaptation (VLMA), which positions a pre-trained VLM as the source model and transfers it towards various unlabel downstream datasets. It proposes a Hough voting-based Self-Training (HoughST) technique that introduces a multimodal Hough voting mechanism to exploit the synergy between vision and language to mitigate the distribution shift in image and text modalities simultaneously. Extensive experiments show that HoughST outperforms the state-of-the-art consistently across 11 image recognition tasks.

**Strengths:**

1.	The paper is well-written and easy to follow.
2.	The paper considers the distribution shifts between pre-training and downstream datasets in VLMs. It points out that the distribution bias arises from both the image and text domains. It is reasonable.
3.	In the experimental section, the proposed method can achieve better performance than existing works. And this paper conducts extensive experiments to validate the effectiveness of proposed visual, text and temporal voting.

**Weaknesses:**

1.	Although the authors emphasize the need to mitigate the visual bias and text bias in VLMs models, the methods used in this paper are quite common and do not contribute new ideas or insights to the field. Specifically, in textural modality, it uses LLM to generate more texts. In visual modality, it uses image augmentation policies. Data augmentation has been widely proven to enhance generalization, so this solution strategy cannot be considered a significant contribution of the article. Considering these points, I think the article lacks sufficient novelty to be accepted. In fact, the work [4] attempts to address visual and text biases in VLMs through a training-free and label-free feature calibration strategy. I suggest the authors can extend their approaches to a training-free strategy to increase its novelty.

2.	Recently, Test-time adaptation (TTA) methods for VLMs have been proposed to mitigate the detrimental impact of the distribution shifts. Compared to unsupervised finetuing, TTA does not require training and quickly updates online based on the test samples. Therefore, could TTA be a more effective strategy for unsupervised vision-language model adaptation? The authors should  discuss the potential tradeoffs between their proposed unsupervised fine-tuning approach and TTA methods, including  computational efficiency, performance, and applicability to different scenarios.

3.	More methods should be compared. Such as TTA methods [1-3]. Can you explain why these specific TTA methods were not included in the comparisons, and discuss how including them might impact the evaluation of their proposed method's effectiveness?

[1] Test-time Prompt Tuning for Zero-shot Generalization in Vision-Language Models. NeurIPS 2022.

[2] C-TPT: Calibrated Test-Time Prompt Tuning for Vision-Language Models via Text Feature Dispersion. ICLR 2024.

[3] Test-time Adaptation with CLIP Reward for Zero-shot Generalization in Vision-Language Models. ICLR 2024.

[4] UMFC: Unsupervised Multi-Domain Feature Calibration for Vision-Language Models. NeurIPS 2024.

**Questions:**

1. Compared  to unsupervised finetuing, TTA  (Test-Time Adaptation) does not require training and quickly updates online based on the test samples. Therefore, could TTA be a more effective strategy for unsupervised vision-language model adaptation? Could you discuss the advantages and disadvantages of unsupervised strategies and TTA for Visual Language Models (VLM)?
2. The performance of TTA methods should be reported and compared.

[1] Test-time Prompt Tuning for Zero-shot Generalization in Vision-Language Models. NeurIPS 2022.

[2] C-TPT: Calibrated Test-Time Prompt Tuning for Vision-Language Models via Text Feature Dispersion. ICLR 2024.

[3] Test-time Adaptation with CLIP Reward for Zero-shot Generalization in Vision-Language Models. ICLR 2024.

---

### Official Review · Reviewer_PmHm · 2024-11-02

**Soundness:** 3
**Presentation:** 3
**Contribution:** 2
**Rating:** 3
**Confidence:** 4

**Summary:**

This paper introduces a method called HoughST for domain adaptation of the vision-language model (especially the CLIP-style models). The goal of this paper is to enhance the performance of CLIP models on downstream datasets usually used for evaluating domain adaptation methods like office-31, office-home, visda, etc. HoughST adopts the idea of voting, augmenting the visual and textual features by gathering features of augmented data. Through the proposed voting technique, they obtained better features which is used to generated pseudo-labels. Pseudo-labels are then used to finetune the VLMs. According to the reported results, this paper have shown effective performance on several datasets.

**Strengths:**

1. The method is technically sound. The augmentation strategy makes sense in improving the feature quality.

2. Compared with baseline CLIP, the method has achieved consistent performance gain.

**Weaknesses:**

What is special about Hough voting? The explanation provided is that it "detects a complex object by voting from its subregion information." Why is this voting method superior to other strategies such as uniform voting, importance-aware voting, weighted voting, and majority voting? Specifically, why is Hough voting more suitable for adapting Vision-Language Models (VLMs)?

Ablation Study Insights: According to the ablation study results in Table 6, it appears that the ST component is more critical for performance improvement. This observation ties back to my first question: why is Hough voting considered special and essential if its performance impact is not significant?

Comparison with SHOT: The procedure of Hough voting resembles the iterative centroids approach used in SHOT. Therefore, what unique element does Hough voting bring to VLMs?

Baseline Comparisons: Although the performance gain against CLIP is evident, the baseline comparison methods seem outdated. The most recent method cited is MUST, proposed in 2022.

Novelty Concerns: The novelty of this method is questionable. Hough voting appears similar to the clustering approach in SHOT, and temporal Hough voting resembles the Exponential Moving Average (EMA) technique.

**Questions:**

Some are in the Weaknesses.

---

### Note · Authors · 2024-11-14

I have read and agree with the venue's withdrawal policy on behalf of myself and my co-authors.